# Identifying opportunity, capability and motivation of Sri Lankan 5th grade schoolteachers to implement in-classroom physical activity breaks: A qualitative study

D. L. I. H. K. Peiris[1], Yanping Duan[1]*, Corneel Vandelanotte[2], Wei Liang[3], Julien Steven Baker[1]

1 Faculty of Social Sciences, Department of Sport, Physical Education and Health, Hong Kong Baptist University, Hong Kong, China, 2 Physical Activity Research Group, School of Health, Medical and Applied Sciences, Central Queensland University, Rockhampton, Australia, 3 College of Physical Education, Shenzhen University, Shenzhen, China

* duanyp@hkbu.edu.hk

**Data Availability Statement:** All relevant, de-identified data are within the paper and its Supporting Information files.

## Abstract

### Background

Classroom-based physical activity interventions have demonstrated positive effects in reducing sedentary behaviour among school children. However, this is an understudied area, especially in low- and middle-income countries such as Sri Lanka. This study aims to explore teachers' opportunity, capability and motivation relating to the implementation of an in-classroom physical activity breaks programme.

### Methods

Twenty-seven teachers were recruited through snowball sampling and participated in semi-structured telephone interviews from early-January to the mid-June 2022. The Capability, Opportunity, and Motivation Behaviour (COM-B) model was used to guide and deductively thematic analyse the interviews.

### Results

21 out of the recruited teachers responded to the full study. The mean age of respondents was 39.24 years old ranging from 27 years to 53 years. Teaching experience of the respondents ranged from three to 37 years, and 57% were female. Three teachers had a degree with a teacher training diploma, while others were having General Certificate of Education in Advanced Level with a teacher training diploma as the highest education qualification. Capability factors such as age, dress code, mask wearing, knowledge, skills and workload of the teachers were identified as important factors in implementing a physical activity breaks intervention in a Sri Lankan classroom setting. Classroom space, facilities, student backgrounds and safety were identified as opportunity factors. Obtaining policy level

**Funding:** The author(s) received no specific funding for this work.

**Competing interests:** The authors have declared that no competing interests exist.

decisions to implement the activity breaks and managing the time of the activities to reduce time lost in education time were identified as motivational factors.

## Conclusion

During the intervention development phase, implementation facilitators and barriers must be considered carefully. Behaviour change techniques can be utilised to address the identified COM-B factors to ensure a good implementation of the intervention.

## Introduction

Of all primary school children in government schools in Sri Lanka, high levels of sedentary behaviour are most prevalent among the fifth graders [1]. The grade five curriculum is identified as the most loaded and competitive when it comes to teaching and learning [2–4]. A contributing factor is that grade five students are expected to sit for a national level competitive scholarship examination in addition to their curriculum-related examinations during the term tests [2–5]. This leads to increased seated learning hours in the fifth-grade teaching. Therefore, there is a need for practically feasible evidence-based interventions to encourage physical activity during classroom time and reduce sedentary behaviour.

In many countries, in-classroom physical activity breaks (IcPAB) interventions have become popular and widely accepted to accommodate the need for reducing seated learning behaviour and improving academic and health-related outcomes in school children. IcPAB have been integrated with subjects such as mathematics [6,7], reading [6], and foreign language learning [8,9], and have demonstrated to reduced seated learning time and increased moderate-to-vigorous physical activity [10,11] and step counts within classrooms. Review studies [12–16] also demonstrate improved cognitive development, anxiety management, mathematics achievement and linguistic literacy achievements. However, to the best of our knowledge IcPAB interventions have not yet been examined in Sri Lankan primary school students.

The most effective health behaviour change interventions are those underpinned by a theoretical basis [13,17]. However, a recent study reviewed ten IcPAB randomised controlled trials exclusively conducted inside primary grade classrooms [13] and indicated that only 20% of the studies were backed by theoretical underpinnings. Hence, to improve the effectiveness of IcPAB interventions they should integrate behaviour change theory during their development [13,17,18]. Recently, several studies [6,8,9] indicated that the Capability, Opportunity, and Motivation Behaviour (COM-B) Model [19,20] can be applied to guide the development of IcPAB interventions due to its sound theoretical basis [19,20] and practicability in applying specific Behaviour Change Techniques (BCTs; [19,20]). The COM-B model is a psychological model of behaviour [21], which was developed by Michie et al. (2011, 2019) as a model embedded into the Behaviour Change Wheel (BCW) framework [19,20]. COM-B model is identified as an effective component in identifying the causes behind a behaviour [22]. The model argues that for a person to engage in an introduced **B**ehaviour (B), he/she must be physically and psychologically be **C**apable (C) of using the existing or introduced social and physical **O**pportunities (O) with sufficient reflective or automatic **M**otivation (M).

Our overarching research project aims to address high seated learning and low physical activity in Sri Lankan primary school students, through the development and evaluation of an IcPAB intervention. In doing so it is essential that any newly developed intervention adheres

to the principles of co-creation [23,24] and is underpinned by a robust behaviour change theory [13]. As such, the specific aims for this study were to explore Sri Lankan primary schoolteachers' opportunity, capability and motivation relating to the implementation of an in-classroom physical activity breaks programme guided by the COM-B Model.

## Methods

The Standards for Reporting Qualitative Research (SRQR; [25]) were used to ensure the quality of the reporting and assessment process [25–28] of this qualitative study. The SRQR checklist is attached in S1 File.

### Qualitative approach and research paradigm

The assessment methods were designed according to the phenomenological research approach, and the interpretivist research paradigm [29,30]. The nature of this study would lead to an understanding of the described opinions and experiences [31] of the suggested concept called in-classroom physical activity breaks for fifth graders in Sri Lankan government schools.

### Researchers' characteristics and reflexivity

The principal investigator, and a research assistant in Sri Lanka conducted the telephone interviews. The interviewers had hands-on experiences in the education system in Sri Lanka, not only as a government university teacher but also as a facilitator who worked with government schools, children, teachers, and other stakeholders. Because of the constructivist nature of this qualitative study, the research team's role was to explore the attitudes and the practice of the target group from the inside based on the culture they live in. This was possible for the interviewers given their deep understanding of the teaching and learning culture of the Sri Lankan government education system.

However, before carrying out the study, the interview guide (S2 File) was discussed among all the team members and full pilot interviews were conducted with six teachers. After the pilot tests, the research team had in-depth discussions of relevant aspects, and did some adaptations to the tone of the questions to be asked in the study. Further to ensure the reflexivity of the study a diary was maintained to note the reflections that occurred during the interview process, biographical accounts of the participants, and troublesome feelings that would occur during the interview process.

### Context

Due to the COVID-19 pandemic, severe social distancing measures were followed by the Sri Lankan government. Hence, phone interviews were conducted with fifth grade teachers from eight out of nine provinces in Sri Lanka (S3 Fig 1 in S3 File). It is recommended to conduct pilot interviews to understand and evaluate the interviewer's weaknesses and strengths [32]. This process refined the interview guide and the schedule. Thus, pilot interviews (n = 6 teachers) were carried out from early-January to the mid-February 2022. The full study (n = 21 teachers participated; n = 1 teacher declined to participate; n = 7 teachers did not respond) was conducted from mid-April to mid-June in 2022. When the interviews were conducted, teachers were conducting both online and physical classes to the elementary school children in response to the change of COVID-19 related restrictions. Thus, the interview timeframe and the associated setting-specific experiences with COVID-19 may have affected teachers' responses to the interview questions.

## Sampling strategy

The first two participants for the interviews were purposively contacted by the interviewers using the principal investigator's personal network. Then the snowballing sample method was used to expand the number of participants in the study. The interviewed teachers were required to speak in Sinhala language and teach fifth grade students in government schools. There was no established relationship between the interviewers and the participants. The teachers were provided with a detailed explanation about the studies aims and encouraged to speak freely of their perceptions.

## Ethical issues pertaining to the human subjects

This study formed part of an intervention development process (Trial registration ID: ISRCTN52180050) and, received ethical approval from the Ethics Review Committee of University of Kelaniya, Sri Lanka (Ref: UOK/ERC/SS/2022/009). Interviewees were guaranteed that the content of the interviews could not be traced back to them during the publication process by confirming the anonymity of the information. Informed verbal consent was obtained from the interviewees prior to the commencement of the interviews.

## Data collection methods

Marshall and Rossman introduced four types of interviews for the interview approach: (1) the informal, conversational interview; (2) the interview guide or topical approach; (3) the standardised, open-ended interview; and (4) the co-constructed or the dialogic interview. This study used the interview guide or the topical approach [33,34] as the interviews were more structured than the informal interviews. Teachers were contacted in the evenings after regular school hours or at weekends. Interview guide was developed in a semi-structured manner with the aim of letting interviewees to orate the responses in-depth and for new topics to emerge. The occurrence of new themes with each new interview was assessed by allowing the research team to evaluate if a saturation of themes occurred. "Such saturation assessment has been proposed as a way of estimating the sample size for qualitative interviews" [35]. After the seventh interview data saturation occurred. However, the research team continued to contact more participants to obtain a wider representativeness from all provinces in Sri Lanka.

## Data collection instrument and technologies

The interview guide was originally developed by the principal investigator and then was refined after discussion with other team members. The interview questions were organised in categories such as current strategies to improve academic achievement, movement behaviours and health enhancement; critical aspects that a prospective IcPAB intervention needs to focus on with special reference to the mathematics and reading achievement, movement behaviours, and health outcomes; and possible challenges of implementing the prospective intervention [36]. Demographic information such as residency, years of teaching experience, the highest education qualification were obtained.

   The perspectives obtained through the interviews were uncovered using content analysis [37,38] and participant-based perspectives [36], to provide insights into the capabilities, opportunities and motivational behaviours of grade five teachers in implanting an IcPAB programme. Many domains were reflected by more than one question, with considerable overlap between them, i.e., respondents' answers may or may not have touched all possible domains associated with a specific question, allowing freedom for in-depth exploration of subjectively relevant topics. The interviews averaged 27 minutes per participant.

### Data processing, data analysis and techniques to enhance trustworthiness

The audio recordings were transcribed verbatim by two research assistants who were fluent in the interviewees' language. The assistants received only sections of the audio recordings for the transliteration. The aim was to ensure deidentification, safety, and integrity during the data processing. All the transcribed documents were password protected further to ensure the confidentiality of the informants and the data. The principal investigator, who was also fluent in the interviewees' language, cross-checked the accuracy of the transcripts with the audio recordings.

The data were coded by the principal investigator. QDA Miner Lite software by Provalis Research, Canada was used for the data analysis. Deductive thematic data analyses [22,39,40] were used as the data analysing technique [22]. The COM-B Model was used as the deductive thematic framework [22,35] for the data analysis to explore teachers' perceptions on enablers/challenges and possible solutions in implementing an IcPAB programme for fifth graders in Sri Lankan government schools. In line with the COM-B Model, the deductive sub-themes were (1) physical and psychological ability to implement an IcPAB, (2) physical and social environment opportunities that would facilitate or hinder the implementation of an IcPAB, and (3) automatic and reflective mechanisms that activate or inhibit the implementation of an IcPAB [21,41–46] Example quotes representing each category within the COM-B components are presented to facilitate the synthesis and interpretation of the qualitative analysis in the results section. Participant quotes are identified by pseudonym, gender and age.

The coding and themes were double-checked by DYP and WL to ensure the trustworthiness and credibility of the data analysis. Disagreements did not occur at this stage. However, minor opinions regarding the analysis were discussed and resolved in meetings that involved the research team. Following this, themes and quotes were translated to English for publication by the principal investigator and double-checked using back translation by a research assistant.

## Results

### Sample characteristics

Twenty-one teachers participated in the full study. One teacher declined to participate, and seven teachers did not respond to the phone calls. The mean age of respondents was 39.24 years old ranging from 27 years to 53 years. Teaching experience of the respondents ranged from three to 37 years (S3 Fig 2 in S3 File), and 57% were female (S3 Fig 3 in S3 File). Three teachers had a degree with a teacher training diploma, while others were having General Certificate of Education (G.C.E.) Advanced Level with a teacher training diploma as the highest education qualification (S3 Fig 4 in S3 File).

### Capability

Analysis of the interview data identified COM-B components, which may inform the development and implementation of an IcPAB programme for grade five students in Sri Lanka. The coding frequency is shown in Fig 1. The physical capability factors such as age, dress code, and mask wearing rules due to COVID-19, can be important for some teachers, while psychological capability was influenced by teachers' confidence on their knowledge, skills and workload regarding the implementation of the IcPAB.

**Physical capability.**   Some teachers expressed their concerns that IcPAB will not "be an easy thing to implement unless the content is physically feasible for teachers who can be

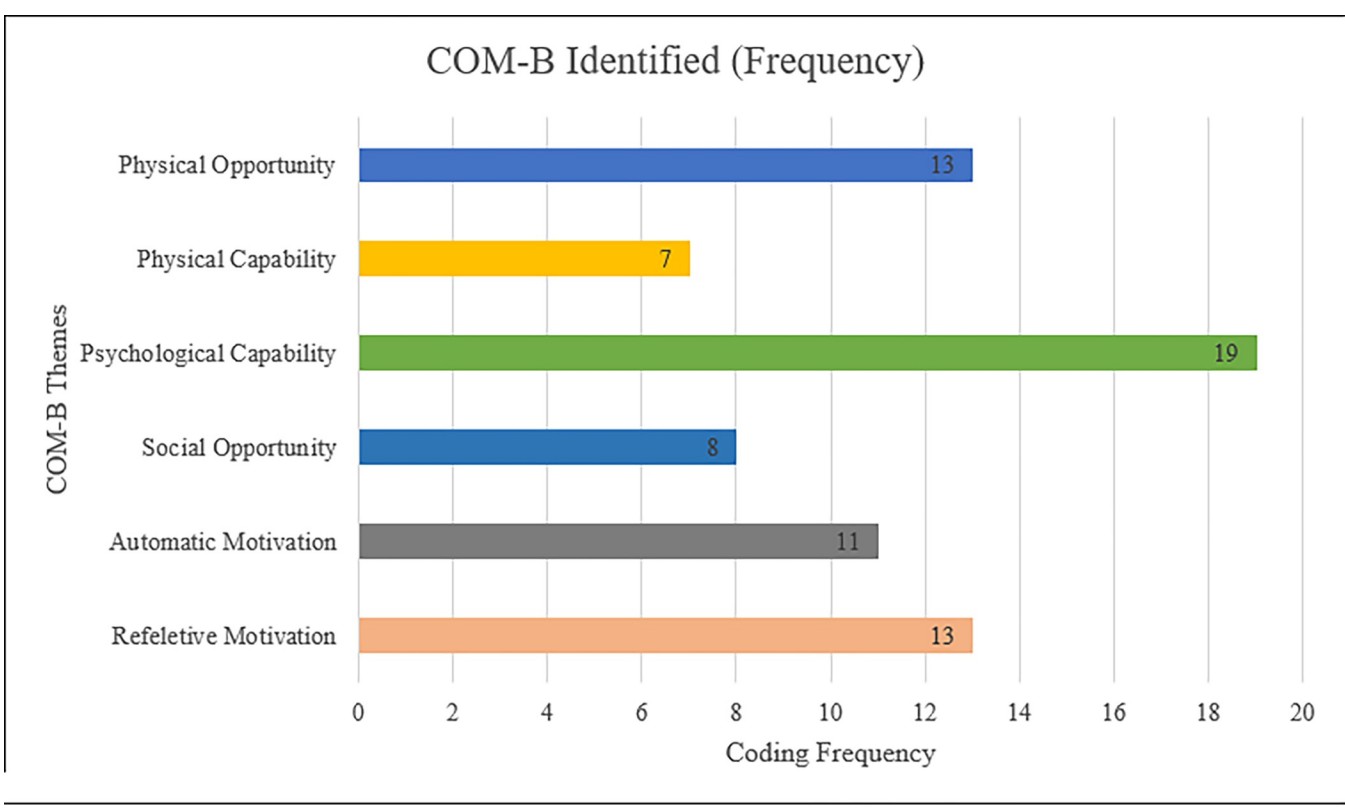

**Fig 1. Coding frequencies of the COM-B Components.**

relatively old" [I1, Female, 45 years], because "there are teachers around their fifties, and they have some physical barriers such as overweight, knee injuries, and fatigue" [I1, Female, 45 years].

> "... when a teacher gets a little older, it is difficult for them to get up and work. Then they think that it would be better if we focused on more seated work... Now I am forty-seven years old. I took the appointment in 2006. It is not like that I cannot do physical activities with aging. But when you get several fourth or fifth grade classes in a row, you get tired. It can happen." [I4, Female, 47 years].

Hence, a teacher also suggested to think about the age-based physical capabilities of teachers when suggesting IcPAB for them to implement.

> "... there are young teachers. There are old teachers. Therefore, it would be good if the physical activities that you recommend us to do in the class are easy for a teacher of all levels to understand and do..." [I3, Female, 28 years].

In addition, some female teachers reflected that the "physical activity breaks would be hard to demonstrate in the classroom as they are wearing 'sarees' every day, which is not a friendly dress for sport like activities" [I12, Female, 28 years], suggesting that physical capability may be a barrier as IcPAB requires physical skills for teachers such as strength and stamina.

Teachers also raised their concerns over the COVID-19 related precautionary behaviours. A teacher said:

*"If you are going to introduce physical activities, think about how they can do those with the masks on... The activities should be designed by considering the social-distancing measures as well. Even though it is hard for us and the students to maintain one meter distance rule, we are always wearing masks."* [I2, Female, 28 years].

**Psychological capability.** Despite teachers' physical capabilities as a barrier to implement an IcPAB programme, psychological capability factors such a knowledge and skills to engaging in physical activity breaks were viewed more positively. Most of the teachers explained that they are confident with the required knowledge and skills to implement an IcPAB as they are "trained teachers to implement such projects once the required information and training is given" [I11, Male, 35 years].

However, the workload of the teachers seemed to be a burden for the teachers.

*"... children are under a lot of pressure. It is not an exam that children willingly face...There are times when the teachers put a lot of pressure on these children. We must cover syllabus... at the same time need to do extra classes prepare them for the scholarship examination... Because of that pressure, students rarely do physical activities inside the classroom... also in the school... Therefore, doing activities in a classroom as you mentioned will be challenging if it takes lots of time from our teaching time..."* [I17, Female, 51 years].

Therefore, the interview data revealed that capability factors must be given major attention when designing IcPAB activities before the implementation stage.

## Opportunity

**Physical opportunity.** Majority of the teachers mentioned that the available space inside a classroom will be a barrier to implementation of IcPAB.

*"... you should always think about the amount of space... Because in many schools, after placing 40 or more students in the class, with tables and chairs ... the actual space is packed. So, there is very little opportunity to do physical activities in the classroom. Therefore, you should make sure that the activities that you suggest can be done without bumping into the body of others... I mean it would be good if the teacher gets an assessment and confidence that they can do it. Because in the end, if the children go to do the said activities and bump into others, fall to the ground, injure their legs, head, or hands, it is very difficult to deal with those problems in a school"* [I11, Male, 35 years].

Even though the classroom space was perceived as a barrier, opportunities to introduce the physical activity breaks inside a classroom were identified. Because teachers mentioned that implementing activities outside the classroom in a regular manner can also become a challenge.

*"... around three hours and twenty minutes per week have been reserved for physical fitness. But there is no way that it is usually done all the time. We use that time sometimes for the morning assembly, to preach the religion... and when it rains or experience bad weather, we cancel outdoor activities... when there are problems... physical active lessons done less. Because we can't take the children outside on a rainy day or a foggy day. Now, especially in a region like ours, it is too foggy in the morning anyway. There are many such problems."* [I2, Female, *28 years*].

Teachers also reported that the consideration of available tangible resources, and the location of the schools, would weigh on the opportunity to introduce an IcPAB.

"*Think of being creative to introduce your project. For example, if you tell us that we need more technology, if you tell us to use computers or use something like a television to do activities with children, that kind of thing is not suitable for all schools. . . many schools in Sri Lanka do not have the resources to try as much as possible. Therefore, it would be good if the activities were introduced to us in such way that we can do them with limited physical resources available in a classroom. . . even in a rural school with the least resources.*" [I7, Female, 36 years].

Interviewees mentioned that a prospective IcPAB programme should be mindful of the participation opportunities for all kinds of children while considering their schools' locations and safety:

"*For example, I used to work in. . . That school was situated in a very difficult location. Children came to school three or four kilometres on foot, jumping over rocks and running through the tea plantations. Such a child does not need any more physical activity. They were also cutting paddy with fathers after school. But in this school, the situation is different. . . Children, directly get out of the car, right near the school gate. . . not even walking a kilometre. So that child needs a lot of activity. Then it is not very practical to give the same activity break to a rural child and an urban child. May be some might have special needs or health issues too. . . it would be better to think of the safety of the kids as well.*" [I6, Male, 40 years].

**Social opportunity.** Teachers mentioned that it is important to receiving direction from authoritative external parties and that there isn't much time to implement the IcPAB:

"*If a programme like this is recommended at the ministry level, or from the regional office itself. . . that means if the officials supervise whether this programme will be implemented, teachers are more likely to do it because of the monitoring. . . if you want to do this work within the expected time, it would be good if we were constantly reminded. Because with the work we have, sometimes we can do those activities for a day or two. But we will go back to our usual routine after that. Therefore, I suggest that either higher-ups occasionally come to school and chat with the teachers and give us the support we need, it is a good thing to make the project successful.*" [I7, Female, 36 years].

"*Anyway, the heavy syllabus should be covered. Two months before the exam, really! there is pressure on the children. Then the children always ask to go to the toilet every ten minutes. Asking to drink water, asking to fill water. Actually, it is not that they want to go to the toilet. They actually want to go out for few minutes. May be if you can use that little minutes to give the students a refreshment that would be great. . . But please, do not take ten, twenty minutes for an activity. . .*" [I10, Female, 27 years].

Teachers also pointed out that ". . .it would be better to have a monitoring system to make sure that the activities are done every day. . . May be not only a record book? Otherwise, I can just mark that I did this today. Only if my heart is honest, I cannot lie. Otherwise, no one knows. Even if I didn't do it, I could mark that I did" [I5, Male, 38 years]. Also, the researchers found that "providing clear guidelines about the activities and having an informed decision at least by the principal to use the physical activities breaks in classes" [I15, Male, 38 years] would ensure the continuity of a IcPAB programme.

## Motivation

It was found that both reflective and automatic motivation play a role in determining the likelihood of teachers carrying out behavioural planning to implement IcPAB.

**Reflective motivation.**   In terms of reflective motivation, many respondents reported that if they believed the IcPAB are a fruitful thing to do then they would be more likely to do it. This was mediated by capability and opportunity components in knowing why the teachers should implement an IcPAB amid a tight workload.

"... *if we have an understanding from some of the things we hear and see, that not only children but anyone who is active can get good physical benefits. However, with the tight syllabus, especially with this scholarship exam, it is difficult to constantly involve those children to do physical activities with the many duties assigned to us... may be if you convince the teachers...if a child is completely active for a certain amount of time a day... then their intelligence will really grow, or confirm that the child will pass the scholarship because of that... I mean, anything about seated time and education or... with acceptable evidence... If there is an understanding at the national level about your programme... it will be a reason for the success of the programme you are trying to bring...*" [I18, Male, 35 years]

"*When the students are getting closer to the scholarship examination date, they seem really stressed or panicked. If the activities that you are going to introduce will ease their psychological burden and help them to learn well, I personally am motivated to implement those activities. I mean at least the student will be less lethargic and smile often...*" [I1, Female, *45 years*]

**Automatic motivation.**   In terms of automatic motivation involving habits, some teachers reported the need for ongoing reinforcement to continue to implement the IcPAB:

"*There was this doctor... When he comes and meets Grade Five teachers in my previous school, he taught us a lot. After I came to this school, I did not meet that doctor. He said...give them the opportunity to draw more pictures, talk to them more, keep them in the front rows and do not blame them if they did not bring a book. I mean we would follow that advice for a week or few days... when time passes, we forget the advice...we become our old selves again. It is neither the teacher's fault nor the child's fault. It is the workload. Therefore, we need regular push with information or reinforcement to motivate ourselves...to help the students to experience happiness when they learn.*" [I6, Male, 40 years].

Furthermore, some teachers also expressed their keenness to implement IcPAB:

"*No problem, I like what you are going to do. Because kids do not like to sit still... need to jump up and shout. That means they are happy. We also will feel a difference may be for few minutes. Because ... if you allocate forty minutes for one subject, if there are forty students in the class, the time that can be given to one child is one minute....*" [I9, Male, 35 years].

## Discussion

This study formed part of an intervention development process that explored the capabilities, opportunities and motivations in relation to classroom-based physical activity breaks interventions (IcPAB) in fifth grade government schoolteachers in Sri Lanka. The sample covered almost all the provinces in Sri Lanka. Despite the heterogeneity of the sample, a considerable

consistency was found in the responses form the interviewees. COM-B model based deductive thematic analysis helped the researchers to identify barriers and facilitators that influence the design and implementation of an IcPAB intervention.

Previous studies did not identify how physiological barriers of teachers can influence the implementation of classroom-based physical activity breaks [17,18]. Nevertheless, physical capability of the teachers such as age, dress code, and mask wearing rules due to COVID-19 might hinder the successful implementation of IcPAB interventions. However, behaviour change techniques can be applied to address the physical capability barriers of the teachers [20,32,47]. For example, similar to previous intervention practices [17,18,48], the intervention design may provide credible written materials and videos of how previous IcPAB practices are conducted despite teachers' age and dress (BCT 4.2—information about antecedents; BCT 5.3 —information about social and environmental consequence; and BCT 9.1—credible source). The teachers need to be taught that the IcPAB can be implemented without changing into a sporty dress by demonstrating the activities in a real class (BCT 6.1—demonstration of the behaviour) as well as by informing them that they can successfully conduct IcPAB frequently (BCT 15.1—verbal persuasion about capability).

Previous studies [18,49] also indicate that the psychological capability of the teachers can boost the successful implementation of the physical activity breaks in a classroom setting. Thus, to increase teacher confidence in implementing the proposed IcPAB programme by managing their workload, techniques such as instruction on how to perform the behaviour (BCT 4.1) in addition to the previously mentioned BCTs (4.2, 6.1, 9.1, 15.1) can be utilised. The intervention designers should (a) provide a manual (33) specifically designed for Sri Lankan teachers with pictures, and videos by merging the curricular with the physical activity breaks; (b) hold special training sessions to advice and agree on the way of performing IcPAB [17,18]; (c) implement IcPAB with teachers/ provide assisted delivery [17]; (d) provide weekly feedback to the teachers and (e) convince that they can successfully conduct the introduced IcPAB through verbal persuasion about capability.

Similar to previous studies [6,17,18,50], physical opportunity was found to be a major influencing factor in determining the content of the IcPAB. Sri Lankan teachers were worried about the space constraints, similar to the teachers in other countries [6,51,52]. Hence, future interventions should focus on showing evidence to the teachers in a real class situation (BCT6.1 and 9.1) and illustrate how IcPAB can be done just by standing behind the desk while designing the content to be matched for a smaller space. Furthermore, the activities should be flexible enough for the teachers to moderate based on the nature of the classroom's amenities, student backgrounds and safety concerns. Training sessions should be held (BCT 6.1) to convince the teachers of IcPAB flexibility as well as its easiness to ensure students' safety.

The need for monitoring and receiving recommendations from higher-ups to implement the intervention was identified as an important social opportunity factor. A recent systematic review [13] also commended the impact of careful monitoring and policy level decisions to increase physically active learning inside classrooms. Hence, the intervention implementers should (a) initiate meetings and request permission from section heads to implement IcPAB as a novel teaching strategy [6], (b) test its effectiveness on academic achievement (BCT 1.2 – problem solving); (c) set goals to achieve a minimum number of IcPAB daily (BCT 1.1 Goal setting (behaviour)) [6]; (d) send reminders to the teachers to implement recommended minimum amount of IcPAB (BCT 7.1 –prompts/ cues) [6] and (e) monitor the way the teachers conduct IcPAB while providing informative evaluative feedback (BCT 2.2 –feedback on behaviour) [6,18].

Reflective and automatic motivation appeared to mediate the capability and opportunity components, similar to previous research [35,53]. The team identified that the teachers should

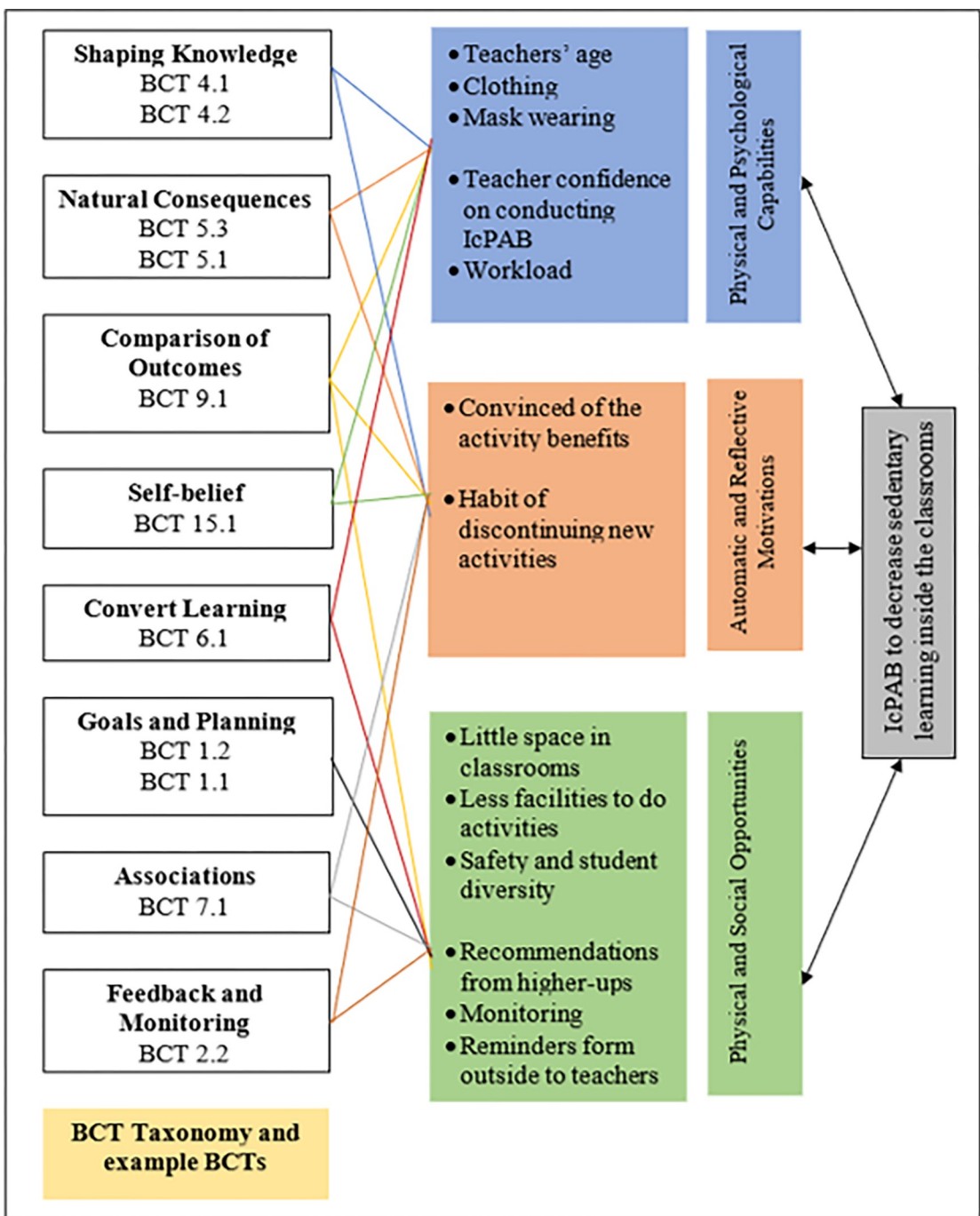

**Fig 2. Reducing sedentary learning through IcPAB programmme among fifth graders linked to the COM-B model and mapped to the BCTv1 (figure adapted from [45,54]).**

be convinced of the benefits of IcPAB to motivate them participate in the intervention. To ensure that teachers believe in the benefits of IcPAB, credible written materials and videos on the risks of prolonged sitting and how such risks can be minimised need to be provided (BCT 5.1—information about health consequences; BCT 5.3 -Information about social and environmental consequence and BCT 9.1), as well as discussing credible evidence from previous

research findings on how IcPAB were helpful in enhancing student performance in various subjects (BCT 4.2, 5.3 and 9.1). The interviews revealed that the teachers are willing to try out the intervention, but they also indicated that they would discontinue to implement IcPAB if they do not receive a continued push to do so. Therefore, in line with previous intervention practices [6,18,48,54,55] strategies such as providing constant reminders (BCT 7.1), and verbally or tangibly reinforcing (positive reinforcements) the teachers (BCT 2.2 and 15.1) who would implement IcPAB should be used. To summarise all of the above, **Fig 2.** illustrates factors influencing the design and implementation of IcPAB according to the COM-B model and mapped with example BCTs.

## Strengths and limitations

Given the nature of the qualitative research methodologies caution should be taken when interpreting the study findings, as they are unlikely to have broad representativeness. However, it is a strength of this study that teachers of nearly every Sri Lankan province participated in the research, and that saturation of themes was reached early in the study.

The use of a widely supported and easily interpreted theoretical model (COM-B) as the basis for this study, is also a strength [21]. We also discussed how the explored COM-B components can be utilised through BCT taxonomy to design and implement an IcPAB intervention for Sri Lankan fifth graders, which further increases the applicability of the findings. Because BCTs provide a useful common language that can be used to mitigate barriers of an intervention.

Despite the limitations, this study yielded detailed data on teachers' opportunities, capabilities and motivations in relation to a classroom-based physical activity programme. Thus, the insights of the study will allow the intervention designers to understand the key areas that should be given attention as well as the barriers and enablers to implementation of the IcPAB. At the same time, the findings could facilitate the design of a feasible, practical, and localised version of an IcPAB programme by referring to existing IcPAB interventions and the BCTv1.

## Conclusions

Fifth grade teachers who were interviewed expressed their motivation to implement classroom-based physical activities if they can be convinced of the benefits of IcPAB. However, physical capability factors (age, dress code, and the mask wearing rule) and psychological capability factors (teachers' confidence on their knowledge and skills, and workload) need to be addressed at the designing phase of the IcPAB intervention. Furthermore, physical opportunities such as teacher apprehensions on classrooms' space, resources, location and safety for different types of students must be addressed. Policy level decisions, allocating fair time for the IcPAB while minimising the loss of pure education time and monitoring of intervention implementation will be needed to enable the successful ongoing implementation of IcPAB interventions.

## Supporting information

**S1 File. The Standards for Reporting Qualitative Research (SRQR).**
(DOCX)

**S2 File. The interview guide.**
(DOCX)

**S3 File. Sample characteristics.**
(DOCX)

**S4 File. Quotes extracted from the interviews for data analysis.**
(DOCX)

## Acknowledgments

We are extremely thankful to all the schoolteachers who provided an immense support in sharing their views. We express our thanks to Miss Yang Ming of the Hong Kong Baptist University, who served as an assistant researcher at the design phase of the study. And we acknowledge the support of the Hong Kong PhD Fellowship Scheme to disseminate the research findings of this study.

## Author Contributions

**Conceptualization:** Yanping Duan, Corneel Vandelanotte, Julien Steven Baker.

**Data curation:** D. L. I. H. K. Peiris, Yanping Duan.

**Formal analysis:** D. L. I. H. K. Peiris.

**Investigation:** D. L. I. H. K. Peiris.

**Methodology:** D. L. I. H. K. Peiris, Yanping Duan.

**Project administration:** D. L. I. H. K. Peiris.

**Resources:** D. L. I. H. K. Peiris.

**Software:** D. L. I. H. K. Peiris.

**Supervision:** Yanping Duan, Corneel Vandelanotte, Wei Liang, Julien Steven Baker.

**Writing – original draft:** D. L. I. H. K. Peiris.

**Writing – review & editing:** D. L. I. H. K. Peiris, Yanping Duan, Corneel Vandelanotte, Julien Steven Baker.

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
