## [Decision Letter · Decision Letter 0]

27 Apr 2023

PONE-D-23-07363Identifying opportunity, capability and motivation of Sri Lankan 5th grade schoolteachers to implement in-classroom physical activity breaks: a qualitative study

PLOS ONE

Dear Dr. Duan,

Thank you for submitting your manuscript to PLOS ONE. After careful consideration, we feel that it has merit but does not fully meet PLOS ONE’s publication criteria as it currently stands. Therefore, we invite you to submit a revised version of the manuscript that addresses the points raised during the review process.

We look forward to receiving your revised manuscript.

Kind regards,

Stevo Popovic, Ph.D.

Academic Editor

PLOS ONE

Journal Requirements:

Additional Editor Comments (if provided):

I am so pleased to finalize the first review round and request minor revision from the authors. The authors need to revise the manuscript according to the reviewers requests and submit revised manuscript and letter to the reviewers as soon as possible.

Reviewers' comments:

Reviewer's Responses to Questions

**Comments to the Author**

1. Is the manuscript technically sound, and do the data support the conclusions?

Reviewer #1: Yes

Reviewer #2: Yes

2. Has the statistical analysis been performed appropriately and rigorously? 

Reviewer #1: N/A

Reviewer #2: Yes

3. Have the authors made all data underlying the findings in their manuscript fully available?

Reviewer #1: Yes

Reviewer #2: Yes

4. Is the manuscript presented in an intelligible fashion and written in standard English?

Reviewer #1: Yes

Reviewer #2: Yes

5. Review Comments to the Author

Reviewer #1: It is necessary to precisely define the sample of respondents in the abstract. Number, age, gender...

I like the idea of this study. Based on this idea, it will be possible to construct policies to improve physical activity and to construct tools for assessing and controlling physical activity.

I think it will contribute to the development of science and practice in the future.

Reviewer #2: The topic of the manuscript is excellent and current. The manuscript is very well methodologically written. The introduction is excellently conceived, in accordance with the objective of the research. The methodology is very well applied, the results are in accordance with the data obtained by statistical analysis. Discussion and Conclusion in accordance with the obtained results, with the presented limitations of the study and recommendations.

6. PLOS authors have the option to publish the peer review history of their article (what does this mean?). If published, this will include your full peer review and any attached files.

Reviewer #1: **Yes: **Bojan Masanovic

Reviewer #2: **Yes: **Jovan Gardasevic, University of Montenegro

---

## [Author Response · Author response to Decision Letter 0]

8 Jun 2023

Dear Dr. Stevo Popovic,

Thank you very much for inviting us to submit a revised version of the manuscript addressing the points raised by reviewers. We have carefully worked on the revision based on the comments of reviewers. Below are our responses to the reviewers, and we are happy to submit the revised manuscript herewith. Thank you. 

Response to the Editor

Additional Editor Comments (if provided):

I am so pleased to finalize the first review round and request minor revision from the authors. The authors need to revise the manuscript according to the reviewers requests and submit revised manuscript and letter to the reviewers as soon as possible.

Response: Thank you very much for informing us about the results in the first review round. We are pleased to accept the comments given by the reviewers and revise the manuscript accordingly.

Response to Reviewers

1: Is the manuscript technically sound, and do the data support the conclusions?

a

Reviewer #1: Yes

Reviewer #2: Yes

Response: Thank you very much for your positive comments. 

 

2: Has the statistical analysis been performed appropriately and rigorously?

Reviewer #1: N/A

Reviewer #2: Yes

Response: Thank you very much for your positive comments. 

3: Have the authors made all data underlying the findings in their manuscript fully available?

Reviewer #1: Yes

Reviewer #2: Yes

Response: Thank you very much for your positive comments. 

4: Is the manuscript presented in an intelligible fashion and written in standard English?

Reviewer #1: Yes

Reviewer #2: Yes

Response 1: Thank you very much for your positive comments. 

5: Review Comments to the Author

Reviewer #1: It is necessary to precisely define the sample of respondents in the abstract. Number, age, gender...

I like the idea of this study. Based on this idea, it will be possible to construct policies to improve physical activity and to construct tools for assessing and controlling physical activity.

I think it will contribute to the development of science and practice in the future.

Response: We thank you for your positive comments and suggestion. We have added sample characteristics information to the abstract. It is presented as follows: 

“21 out of the recruited teachers responded to the full study. The mean age of respondents was 39.24 years old ranging from 27 years to 53 years. Teaching experience of the respondents ranged from three to 37 years, and 57% were female. Three teachers had a degree with a teacher training diploma, while others were having General Certificate of Education in Advanced Level with a teacher training diploma as the highest education qualification.”

Reviewer #2: The topic of the manuscript is excellent and current. The manuscript is very well methodologically written. The introduction is excellently conceived, in accordance with the objective of the research. The methodology is very well applied, the results are in accordance with the data obtained by statistical analysis. Discussion and Conclusion in accordance with the obtained results, with the presented limitations of the study and recommendations.

Response: Thank you very much for your positive comments. 

6: PLOS authors have the option to publish the peer review history of their article (what does this mean?). If published, this will include your full peer review and any attached files.

Do you want your identity to be public for this peer review? For information about this choice, including consent withdrawal, please see our Privacy Policy.

Reviewer #1: Yes: Bojan Masanovic

Reviewer #2: Yes: Jovan Gardasevic, University of Montenegro

Response: Thank you for agreeing to make available your information in public. We are honoured to obtain your review of our paper. 

In the case of questions, please do not hesitate to contact me. We are looking forward to your reply.

Sincerely yours

Dr Duan Yanping

Hong Kong Baptist University

Hong Kong 

28th April 2023

---

## [Decision Letter · Decision Letter 1]

7 Jul 2023

Identifying opportunity, capability and motivation of Sri Lankan 5th grade schoolteachers to implement in-classroom physical activity breaks: a qualitative study

PONE-D-23-07363R1

Dear Dr. Duan,

We’re pleased to inform you that your manuscript has been judged scientifically suitable for publication and will be formally accepted for publication once it meets all outstanding technical requirements.

Kind regards,

Stevo Popovic, Ph.D.

Academic Editor

PLOS ONE

Additional Editor Comments (optional):

After the second turn of revision process I decided to accept the manuscript.

Reviewers' comments:

Reviewer's Responses to Questions

**Comments to the Author**

1. If the authors have adequately addressed your comments raised in a previous round of review and you feel that this manuscript is now acceptable for publication, you may indicate that here to bypass the “Comments to the Author” section, enter your conflict of interest statement in the “Confidential to Editor” section, and submit your "Accept" recommendation.

Reviewer #1: All comments have been addressed

Reviewer #2: All comments have been addressed

2. Is the manuscript technically sound, and do the data support the conclusions?

Reviewer #1: Yes

Reviewer #2: Yes

3. Has the statistical analysis been performed appropriately and rigorously? 

Reviewer #1: Yes

Reviewer #2: Yes

4. Have the authors made all data underlying the findings in their manuscript fully available?

Reviewer #1: Yes

Reviewer #2: Yes

5. Is the manuscript presented in an intelligible fashion and written in standard English?

Reviewer #1: Yes

Reviewer #2: Yes

6. Review Comments to the Author

Reviewer #1: (No Response)

Reviewer #2: The topic of the manuscript is excellent and current. The manuscript is very well methodologically written. The introduction is excellently conceived, in accordance with the objective of the research. The methodology is very well applied, the results are in accordance with the data obtained by statistical analysis. Discussion and Conclusion in accordance with the obtained results, with the presented limitations of the study and recommendations.

7. PLOS authors have the option to publish the peer review history of their article (what does this mean?). If published, this will include your full peer review and any attached files.

Reviewer #1: No

Reviewer #2: **Yes: **Jovan Gardasevic, University of Montenegro

---

## [Editor Report · Acceptance letter]

12 Jul 2023

PONE-D-23-07363R1 

Identifying opportunity, capability and motivation of Sri Lankan 5th grade schoolteachers to implement in-classroom physical activity breaks: a qualitative study 

Dear Dr. Duan:

I'm pleased to inform you that your manuscript has been deemed suitable for publication in PLOS ONE. Congratulations! Your manuscript is now with our production department. 

Kind regards, 

on behalf of

Professor Stevo Popovic 

Academic Editor

PLOS ONE